# Healthcare utilization 9 months pre- and post-COVID-19 hospitalization among patients discharged alive

**Mohammed Zaidan**[1‡]*, **Daniel Puebla Neira**[2‡], **Efstathia Polychronopoulou**[3], **Kuo Yong-Fang**[3,4], **Gulshan Sharma**[1]

**1** Department of Internal Medicine, Division of Pulmonary, Critical Care and Sleep Medicine, University of Texas Medical Branch (UTMB), Galveston, TX, United States of America, **2** Department of Internal Medicine, Division of Pulmonary, Critical Care and Sleep Medicine, University of Arizona College of Medicine, Phoenix, AZ, United States of America, **3** Office of Biostatistics, University of Texas Medical Branch (UTMB), Galveston, TX, United States of America, **4** Sealy Center on Aging, University of Texas Medical Branch (UTMB), Galveston, TX, United States of America

‡ Dr. Zaidan and Dr. Puebla Neira are co-first authors and contributed equally to the work in this manuscript.
* mfzaidan@utmb.edu

## Abstract

### Background

Emerging evidence suggests that there is an increase in healthcare utilization (HCU) in patients due to Coronavirus Disease 2019 (COVID-19). We investigated the change in HCU pre and post hospitalization among patients discharged home from COVID-19 hospitalization for up to 9 months of follow up.

### Study design and methods

This retrospective study from a United States cohort used Optum® de-identified Clinformatics Data Mart; it included adults discharged home post hospitalization with primary diagnosis of COVID-19 between April 2020 and March 2021. We evaluated HCU of patients 9 months pre and post -discharge from index hospitalization. We defined HCU as emergency department (ED), inpatient, outpatient (office), rehabilitation/skilled nursing facility (SNF), telemedicine visits, and length of stay, expressed as number of visits per 10,000 person-days.

### Results

We identified 63,161 patients discharged home after COVID-19 hospitalization. The cohort of patients was mostly white (58.8%) and women (53.7%), with mean age 72.4 (SD± 12) years. These patients were significantly more likely to have increased HCU in the 9 months post hospitalization compared to the 9 months prior. Patients had a 47%, 67%, 65%, and 51% increased risk of ED (rate ratio 1.47; 95% CI 1.45–1.49; p < .0001), rehabilitation (rate ratio 1.67; 95% CI 1.61–1.73; p < .0001), office (rate ratio1.65; 95% CI 1.64–1.65; p < .0001), and telemedicine visits (rate ratio 1.5; 95% CI 1.48–1.54; p < .0001), respectively. We also found significantly different rates of HCU for women compared to men (women

**Data Availability Statement:** Optum data is proprietary (and we have purchased access to it under contract), therefore, we are not permitted to share the data. We accessed Clinformatics through

the university contract. This contract expired on 6/30/2023 To acquire the same type of data, researchers need to sign a contract with Optum. Contact information for Optum: call: 1-866-3061321; email: connected@optum.com The cohort and analytic results can be replicated by following the inclusion/exclusion criteria as described in the manuscript. Interested researchers would need to sign individual contracts with Optum and receive the data. We did not have any special access privileges to Optum data.

**Funding:** Dr. Puebla Neira reports support from NHLBI Division of Intramural Research (US), NHLBI Advanced Respiratory Research for Equity (AiRE) - AZ-PRIDE Program grant (5R25HL126140-09) during the conduct of this study. The funders had no role in study design, data collection and analysis, decision to publish, or preparation of the manuscript.

**Competing interests:** The authors have declared that no competing interests exist.

have higher risk of ED, rehabilitation, and telemedicine visits but a lower risk of inpatient visits, length of stay, and office visits than men) and for patients who received care in the intensive care unit (ICU) vs those who did not (ICU patients had increased risk of ED, inpatient, office, and telemedicine visits and longer length of stay but a lower risk of rehabilitation visits). Outpatient (office) visits were the highest healthcare service utilized post discharge (64.5% increase). Finally, the risk of having an outpatient visit to any of the specialties studied significantly increased post discharge. Interestingly, the risk of requiring a visit to pulmonary medicine was the highest amongst the specialties studied (rate ratio 3.35, 95% CI 3.26–3.45, p < .0001).

## Conclusion

HCU was higher after index hospitalization compared to 9 months prior among patients discharged home post-COVID-19 hospitalization. The increases in HCU may be driven by those patients who received care in the ICU.

## Introduction

Most patients are discharged alive from hospitalization due to Coronavirus Disease 2019 (COVID-19). With over 6 million COVID-19 hospitalizations in the United States (US), there is growing concern regarding the health care utilization (HCU) of patients post discharge [1–4]. Prior reports of non-COVID-19 patients show high HCU post hospitalization and post care in an intensive care unit (ICU) [5–8]. Based on this evidence, HCU is expected to be high for patients discharged from COVID-19 admission.

Overall, studies have found that patients who tested positive for severe acute respiratory syndrome coronavirus-2 (SARS-COV-2) have greater HCU than patients who tested negative [9, 10]. Also, the 30-day and 60-day readmission rate for patients discharged alive after a hospitalization due to COVID has been reported as 13% to 24% and 19.9%, respectively [2, 11–17]. Similarly, the rates of ambulatory visits, emergency department (ED) visits, and hospitalizations vary following discharge from COVID-19 in socially disadvantaged patients compared to those who are socially advantaged [18, 19]. It has been reported that 82.1% of patients had follow-up visits with a primary care provider in the 60 days following discharge from COVID-19 hospitalization [20]. However, these observations were obtained from cohorts of patients who were ambulatory and/or post-hospital discharge. The HCU of patients discharged home post-COVID-19 hospitalization is not well understood. We, therefore, investigated the HCU of patients discharged home from COVID-19 hospitalization for 9 months post-discharge, using a national US database. We hypothesized that HCU will be high post discharge and that certain specialties will see a disproportionate increase in their share of post-discharge HCU. Some of the results of this study were previously presented in the form of a conference abstract [21].

## Study design and methods

### Data source

We used de-identified data from Optum's Clinformatics Data Mart (CDM), a database of administrative health claims for members of large commercial and Medicare Advantage health plans. Optum's CDM is a comprehensive database that includes claims from enrollees with either commercial insurance or Medicare Advantage Plans, encompassing over 63 million

unique enrollees across the United States from all states. Notably, more than 95% of the enrollees in Optum's CDM possess commercial insurance. This database does not include traditional Medicare or Medicaid enrollees, implying a potential underrepresentation of the older or low socioeconomic status populations. A significant aspect of the data from Optum's CDM is its source diversity; since the data originates from insurance claims. This inclusivity means data can come from a variety of healthcare settings, whether rural or urban, academic or community-based. Whenever a CDM patient files a claim with their insurance, this information is captured in the database, regardless of the specific medical center or office they visited. This wide-ranging data collection provides a broad and diverse view of healthcare utilization and patterns across different demographics and geographies in the United States. The University of Texas Medical Branch Institutional Review Board (IRB) approved this study (IRB# 20–0180). The need for informed consent was waived by the IRB due to the de-identified nature of the study.

## Cohort selection

Our sample included all adults hospitalized with a primary diagnosis of COVID-19 who were discharged home (with or without home health care) between April 2020 and March 2021, with at least 12 months of continuous enrollment before this diagnosis. COVID-19 cases were identified by the International Classification for Diseases, tenth revision, clinical modification (ICD-10-CM) diagnosis code U07.1 or from a positive test. We excluded patients who were not discharged home or whose insurance coverage ended during the inpatient stay (Fig 1).

## Variables

We collected information on patient demographics: age at time of COVID-19 diagnosis, sex, race/ethnicity, and region of residence. We identified comorbidities by ICD-10-CM diagnosis codes, with a 12-month lookback period before COVID-19 hospitalization.

We evaluated HCU in the 9 months pre and post hospitalization from COVID-19. We divided the study period into pre-COVID-19 hospitalization (-0-9 months) and post-COVID-19 discharge (+0–9 months). We also subdivided into 3-month periods for pre-COVID-19 hospitalization (-0-3 months, -3-6 months, and -6-9 months before hospitalization) and post-COVID-19 discharge (+0–3 months, +3–6 months, and +6–9 months post discharge). HCU included ED visits, inpatient admissions, rehabilitation/skilled nursing facility (SNF) admissions, outpatient visits, and telemedicine visits. Based on billing information, we examined outpatient visits (from here on out referred to as office visits) to primary care providers, including family medicine, internal medicine, and nurse practitioner visits, and certain subspecialties, such as cardiology, pulmonary medicine, endocrinology, neurology, physical medicine and rehabilitation, psychiatry, and other mental health professionals.

## Statistical analysis

Cohort characteristics are presented as mean and standard deviation (SD), median and interquartile range, or frequency and percentage. HCU in the pre-COVID-19 hospitalization and post-COVID-19 discharge periods was expressed as the number of visits per 10,000 (10k) person-days, where each patient contributed person-days until death, loss of insurance eligibility, or the end of the 9-month follow-up period. To identify the risk of HCU post-COVID-19 compared to pre-COVID-19 hospitalization incidence rate ratios (number of observed events over total person-days contributed) and exact Poisson confidence intervals were calculated for each period [22]. All findings were considered statistically significant if the *P* value was <0.05. To address the concern of survivor bias, the study required participants to have at least 12 months

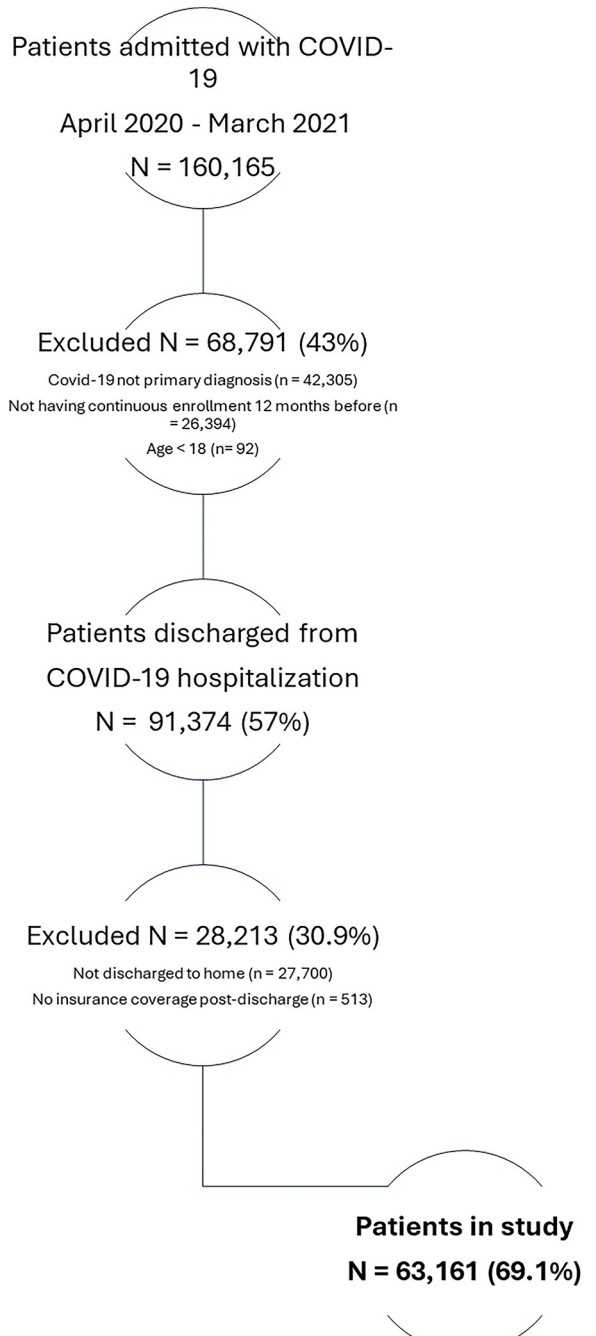

**Fig 1. Cohort selection of patients discharged from COVID-19 hospitalization.** COVID-19 was identified by the ICD-10-CM diagnosis code U07.1 or from a positive test. Patients were excluded if they were not discharged home or if their insurance coverage ended during the inpatient stay. *Abbreviations*: COVID-19: Coronavirus Disease 2019; Optum CDM: Optum's Clinformatics Data Mart. ICD-10-CM: International Classification for Diseases, tenth revision, clinical modification.

of data prior to their primary COVID-related hospitalization for accurate comorbidity assessment, without mandating post-hospitalization follow-up, and utilized a person-days approach to accommodate variable follow-up durations. As a sensitivity analysis, we evaluated HCU post discharge in women and compared it to that of men. In addition, we analyzed HCU in

patients who received care in the ICU and compared it to those who did not. We also examined the distribution of office visits per 10K person-days by provider specialty (select specialties) in each period and calculated rate ratios to identify the risk of having a visit with that specialist post-COVID-19 hospitalization compared to the pre-hospitalization period. All analyses were conducted using SAS version 9.2 (SAS Institute, Inc., Cary NC).

## Results

In this retrospective study, we identified 63,161 patients discharged home after hospitalization due to COVID-19. This cohort was comprised of patients who were mostly white (58.8%), women (53.7%), and with mean age of 72.4 (SD± 12.8) years. The most common comorbidities of these patients were hypertension (75.8%), diabetes mellitus (36.5%), congestive heart failure (25.8%), coronary artery disease (25.2%), and chronic obstructive pulmonary disease (23%) (Table 1).

### Health care utilization after discharge home from hospitalization due to COVID-19

Patients discharged home from COVID-19 hospitalization were significantly more likely to have increased HCU in the 9 months post hospitalization compared to the 9 months prior to such hospitalization (Tables 2, 3, S1-S6, S8 Tables in S1 File). For example, these patients had 47%, 67%, 65%, and 51% increased risk of ED (rate ratio 1.47; 95% CI 1.45–1.49; p < .0001), rehabilitation (rate ratio 1.67; 95% CI 1.61–1.73; p < .0001), office (rate ratio1.65; 95% CI 1.64–1.65; p < .0001), and telemedicine visits (rate ratio 1.5; 95% CI 1.48–1.54; p < .0001), respectively. Also, the post-discharge risk of inpatient visits (rate ratio 2.20; 95% CI 2.14–2.25; p < .0001) and longer length of stay (rate ratio 2.62; 95% CI 2.59–2.64; p < .0001) doubled compared to the pre-COVID-19 hospitalization period (Table 2 and S1 Table in S1 File).

As a sensitivity analysis, we evaluated HCU post discharge in women and compared it to that of men patients (S5 Table in S1 File). We also analyzed HCU in patients who received care in the ICU and compared it to those who did not (S6 Table in S1 File). We found that women have higher risk of ED (rate ratio 1.03; 95% CI 1.01–1.05), rehabilitation (rate ratio 1.23; 95% CI 1.17–1.30), and telemedicine visits (rate ratio 1.12; 95% CI 1.1–1.16) than men. But women have a lower risk of inpatient visits (Rate ratio 0.91, 95% CI 0.88–0.94), and office visits (rate ratio 0.92; 95% CI 0.91–0.93) than men. Additionally, women had a shorter length of stay (rate ratio 0.84, 95% CI 0.84–0.86) (S5 Table in S1 File).

Compared to patients who did not receive ICU care, those who were admitted to the ICU have increased risk of ED visits (19%, rate ratio 1.19; 95% CI 1.17–1.22), inpatient visits (27%, rate ratio 1.27; 95% CI 1.23–1.31); shorter length of stay (34%, rate ratio 1.34; 95% CI 1.32–1.35), office visits (46%, rate ratio 1.46; 95% CI1.44–1.47), and telemedicine visits (38%, rate ratio 1.38; 95%1.35–1.43).

Contrary to the elevated risks stated above, patients discharged home from COVID-19 hospitalization who received care in the ICU were less likely to have rehabilitation visits (rate ratio 0.92; 95% CI 0.87–0.97) compared to those who did not receive care in the ICU (S6 Table in S1 File).

Outpatient office visits were the most utilized health care service by patients post discharge (275.5 visits-10k person-days). All included specialties showed more visits in the 9 months after discharge than in the 9 months prior to hospitalization. Primary care providers had the highest number of visits post discharge (133.1 visits/10k persons-days), followed by cardiology (21.9 visits/10k persons-days) and pulmonary medicine (12.9 visits/10k persons-days). Interestingly, pulmonary medicine saw the highest percent change in the number of pre- (3.8 visits/10k person-days) vs post-discharge visits (235.5% change). Similarly, the risk of having a visit

**Table 1. Characteristics of patients discharged home from COVID-19 hospitalization from April 2020 to March 2021 in the United States.**

| Characteristics | N = 63,161 (100%)[a,b,c] |
|---|---|
| **Age, mean (SD)** | 72.4 (12.8) |
| **Sex** | - |
| Women | 33897 (53.7) |
| Men | 29263 (46.3) |
| **Race/Ethnicity [d,e]** | - |
| White | 37135 (58.8) |
| Hispanic | 9843 (15.6) |
| Black | 10443 (16.5) |
| Asian | 1473 (2.3) |
| Other/Unknown | 4261 (6.8) |
| **Region** | - |
| Midwest | 13897 (22) |
| Northeast | 8465 (13.4) |
| South | 30145 (47.7) |
| West | 9299 (14.7) |
| Unknown | 1355 (2.1) |
| **Comorbidities** | - |
| DM | 23,065 (36.5) |
| HTN | 47,890 (75.8) |
| Asthma | 6,285 (10.0) |
| COPD | 14,534 (23.0) |
| CKD | 6,793 (10.8) |
| ESRD | 2,032 (3.2) |
| Stroke | 5,547 (8.8) |
| CHF | 16,303 (25.8) |
| Cancer | 7,982 (12.6) |
| CAD | 15,925 (25.2) |
| Liver disease | 3,871 (6.1) |
| **ICU use** | 21760 (34.5) |

[a] Our cohort consists of 63,161 patients discharged home after hospitalization due to COVID-19.

[b] COVID-19 and comorbidities were identified by ICD-10-CM diagnosis codes(S7 Table in S1 File).

[c] Patients were excluded if they were not discharged home or if their insurance coverage ended during the inpatient stay.

[d] Patients self-identifying as non-Hispanic ethnicity were categorized based on race (White, Black, Asian, other/unknown).

[e] Patients self-identifying as Hispanic ethnicity were included in the Hispanic group regardless of race.

*Definition of abbreviations*: COVID-19: Coronavirus Disease 2019; SD: Standard Deviation; DM: Diabetes Mellitus; HTN: Hypertension; COPD: Chronic Obstructive Pulmonary Disease; CKD: Chronic Kidney Disease; ESRD: End-Stage Renal Disease; CHF: Congestive Heart Failure; CAD: Coronary Artery Disease; ICU: Intensive Care Unit; ICD-10-CM International Classification for Diseases, tenth revision, clinical modification.

to any specialty significantly increased post discharge, but the risk of having a visit with pulmonary medicine was the highest (rate ratio 3.35, 95% CI 3.26–3.45, p < .0001) (Table 3, S3, S4 Tables in S1 File).

We also found that the risk of having visits with neurology (rate ratio 1.51; 95% CI 1.46–1.58; p < .0001), psychiatry (rate ratio 1.41; 1.35–1.49; p<0.0001) and other mental health

**Table 2. Health care utilization of patients pre and post-hospitalization due to COVID-19[,c,d,e].**

| - | Pre-COVID-19 Hospitalization [a,b] | Post-COVID-19 Hospitalization [a,b] | - | - | - |
|---|---|---|---|---|---|
| - | -0-9 months | +0-9 months | Percent change [c] | Rate ratio (95% CI) | p-value |
| **ED visits** | 22.0 | 32.3 | 47.0 | 1.47 (1.45–1.49) | < .0001 |
| **Inpatient visits** | 7.0 | 15.4 | 119.6 | 2.20 (2.14–2.25) | < .0001 |
| **Inpatient admission (LOS)[d]** | 42.7 | 111.6 | 161.6 | 2.62 (2.59–2.64) | < .0001 |
| **Rehabilitation visits** | 3.3 | 5.5 | 66.7 | 1.67 (1.61–1.73) | < .0001 |
| **Office visits** | 156.5 | 257.5 | 64.5 | 1.65 (1.64–1.65) | < .0001 |
| **Telemedicine visits** | 11.2 | 17.0 | 51.3 | 1.51 (1.48–1.54) | < .0001 |

[a] Our cohort consists of 63,161 patients discharged home after hospitalization due to COVID-19.

[b] Health care utilization of patients, measured by number of visits per 10k person-days.

[c] Percent change of HCU pre- and post- hospitalization due to COVID-19.

[d] LOS was defined as the number of days of inpatient status after hospital admission.

*Definition of abbreviations*: COVID-19: Coronavirus Disease 2019; ED: Emergency Department; ICD-10-CM: International Classification for Diseases, tenth revision, clinical modification; LOS: length of stay. 10k = 10,000

professionals (rate ratio 1.66; 95% CI 1.54–1.79; p<0.0001) significantly increased post discharge compared to the pre-hospitalization period (Table 3).

## Discussion

In our retrospective cohort study of patients discharged home after a COVID-19 hospitalization in the US, we found that these patients have an increased risk of post-discharge HCU compared to the pre-hospitalization period. Our results are similar to studies of HCU in patients post discharge from non-COVID-19 and COVID-19 admissions [3]. Our findings advance the knowledge about HCU post-COVID hospitalization in patients discharged home. Additionally, our observations of different HCU in women compared to men and in those who received care in the ICU compared to those who did not may help health systems and policy makers identify potential disparities and at-risk populations who may benefit from targeted interventions to improve their HCU post discharge. Furthermore, we found that all medical

**Table 3. Health care utilization of patients pre and post hospitalization due to COVID-19[,] by medical specialty.**

| - | Pre-COVID-19 Hospitalization[a,b] | Post-COVID-19 Hospitalization[a,b] | - | - | - |
|---|---|---|---|---|---|
| - | -0-9 months | +0-9 months | Percent change [c] | Rate ratio (95% CI) | p-value |
| **PCP** | 78.8 | 133.1 | 68.9 | 1.69 (1.67–1.70) | < .0001 |
| **Cardiology** | 12.2 | 21.9 | 79.2 | 1.79 (1.76–1.83) | < .0001 |
| **Pulmonary Medicine** | 3.8 | 12.9 | 235.5 | 3.35 (3.26–3.45) | < .0001 |
| **Endocrinology** | 2.3 | 3.7 | 63.8 | 1.64 (1.57–1.71) | < .0001 |
| **Neurology** | 3.0 | 4.5 | 51.6 | 1.51 (1.46–1.58) | < .0001 |
| **Phys Med & Rehab** | 2.0 | 2.5 | 25.8 | 1.26 (1.20–1.32) | < .0001 |
| **Psychiatry** | 1.9 | 2.7 | 41.4 | 1.41 (1.35–1.49) | < .0001 |
| **Mental Health Professional** | 0.8 | 1.3 | 66.1 | 1.66 (1.54–1.79) | < .0001 |

[a] Our cohort consists of 63,161 patients discharged home after hospitalization due to COVID-19.

[b] Health care utilization of patients, measured by number of visits per 10k person-days.

[c] Percent change of HCU from pre- to post-hospitalization periods due to COVID-19.

*Definition of abbreviations*: HCU: health care utilization; COVID-19: Coronavirus Disease 2019; PCP: Primary Care Provider; ICD-10-CM: International Classification for Diseases, tenth revision, clinical modification; Phys Med & Rehab: Physical Medicine and Rehabilitation; 10k = 10,000.

specialties studied had high use that varied by specialty, which can help us identify medical providers needed to meet patient care demands during the next respiratory pandemic.

We must consider why patients discharged home from a hospitalization for severe COVID-19 have high HCU. Acute infection by SARS-COV-2, leading to hospitalization due to severe COVID-19, has been associated with conditions that have multi-organ involvement and dysfunction, most commonly pneumonia but also including cardiac injury, acute liver injury, acute kidney injury, venous and arterial thrombotic events, and a variety of neurological and psychiatric manifestations [23]. Multisystem infection of the virus may explain a variety of persistent organ dysfunctions and may result in chronic clinical symptoms [23, 24]. These persistent symptoms likely lead patients to seek medical care post hospitalization and health systems to develop multi-disciplinary clinics, facilitating referrals to multiple specialists [25]. Our findings of increase in post-discharge HCU suggest an increased demand for hospital-centered care.

The differences in HCU in women compared to men are intriguing. Prior literature showed that women had lower risk of adverse outcomes and mortality due to COVID-19 compared to men [26, 27]. In our study, women have higher risk of ED visits, rehabilitation visits, and telemedicine visits, and lower risk of inpatient visits, office visits compared to men. Women also had a shorter length of stay compared to men. Although we do not know why these differences exist, our findings are consistent with prior non-COVID-19 literature of fewer hospital admissions, shorter length of stay, and fewer physician visits in women compared to men [28, 29]. The differences in HCU by sex may be explained by a variety of factors, including demographics and social factors, such as health care needs (limitation in mobility, disability, specific chronic comorbidities) and economic access factors (overall health, income, education, etc.) [30].

Our finding of higher HCU in patients post-discharge home from COVID-19 may be primarily driven by patients who received care in the ICU. Pre-COVID-19, research showed that patients with sepsis, pneumonia, central line associated blood stream infections, and ventilator associated pneumonia had increased post-discharge mortality and high HCU [8]. Also, patients who receive prolonged mechanical ventilation (>21 days) have high risk of mortality, readmissions to the hospital and ICU, and high HCU [5]. Finally, patients who survive ARDS have impaired functional status, and their quality of life is affected even 2 years after discharge from the ICU [7]. In our cohort, 34.5% of patients received care in the ICU, which is in accordance to published findings that nearly 1 in 3 (33%) hospitalized patients with COVID-19 develop ARDs and 1 in 4 hospitalized patients require transfer to the ICU (26%) [31]. We also know that patients with COVID-19 admitted to the ICU have longer length of stay in the hospital and ICU, longer length of mechanical ventilation, and therefore may be at increased risk of nosocomial infections [32, 33]. If we are to extrapolate those percentages to the over 6 million people hospitalized due to COVID 19, it may be expected that over 1.5 million people may experience the elevated HCU described in our study.

Notable insights emerged from our study regarding medical specialty utilization post-COVID-19 discharge. Primary care providers and outpatient office visits were pivotal, while an intriguing surge in pulmonology clinic visits highlighted disproportionate escalating demand for specialized respiratory care. This may be explained by the persistent pulmonary symptoms and complications post-COVID-19. A recent study highlighted the multisystemic impact of COVID-19 and reported that the chronic symptoms linked with SARS-CoV-2 infection involved a variety of organ systems, with chronic cough notably identified as one of the defining symptoms for the new diagnosis of post-acute sequelae of SARS-CoV-2 infection (PACS) [34]. Another study found that SARS-COV-2 infection was associated with an additional 213 health care visits per 1000 patients during the 6 months after the acute stage of

illness [8]. Notably, this study found the second highest increase in utilization was observed for pulmonary symptoms (bronchitis, venous thromboembolism, dyspnea upon exertion, hypoxemia, and cough). Another study in France evaluated sequelae in COVID-19 patients post-hospital discharge from March to May 2020. They found that 51% of patients reported persistent respiratory symptoms 4 months after COVID-19 hospitalization. This study found persistent ground glass opacities in 32%, fibrotic lung lesions in 12%, and abnormal diffusion capacity in 13.6% of patients 4 months post-discharge [35]. Other studies have reported persistent reductions in diffusion capacity 3 months to 4 months after acute illness, with rates ranging from 16.4% to 52% [4, 36]. Multivariate analysis studies also found that severe disease during acute illness was associated with a persistently reduced diffusion capacity [36] and worse heart function [24]. Additionally, the most frequent serious manifestation of acute COVID-19 infection is pneumonia [37], with a reported 17% of patients complicated with acute respiratory distress syndrome [38]. All of the above may help explain our finding of 235.5% change in pulmonary visits from the pre- to the post-hospitalization period.

Interestingly, studies have reported persistent pulmonary symptoms in patients without persistent physiologic impairments [39, 40]. One study aimed to investigate the long-term pulmonary effects of severe COVID-19 pneumonia by assessing cardiopulmonary exercise test (CPET) performance in 60 patients 12 months after a COVID-19 infection that required ICU management [41]. Exercise capacity assessed by CPET was within normal limits in most patients 12 months after hospitalization, and impairment was predominantly related to persistent deconditioning or prior respiratory comorbidities. Complementing our findings, other studies have compared hospitalization related to COVID to other causes of hospitalization.

The disproportionate increases in HCU by PCP and pulmonary specialist may also be explained by the establishment of multi-disciplinary post-acute sequelae of COVID-19 (PASC) clinics. One study surveyed healthcare systems in the US participating in the PETAL Network and reported that 70% had established an outpatient clinic for PASC with physicians providing care in 97% of clinics supplemented by Advanced Practice Professionals [25]. Of these systems, 21% automatically referred all patients discharged alive post-COVID-19 hospitalization to the PASC clinic for outpatient follow up while 70% of referrals relied on physician discretion or patient requests. Subspecialties available were pulmonary (97%), general medicine and primary care (58%), cardiology (52%), and psychiatry (30%). Remarkably, 73% of these PASC clinics were distinct from their previously established post-ICU clinics [25].

We acknowledge our study's limitations, including its retrospective design; therefore, we cannot infer cause and effect but only an association between being discharged from a hospitalization due to COVID-19 and an increase in post-discharge HCU. Also, our patient cohort was obtained from a national commercially-insured and Medicare Advantage claims database. Therefore, we do not provide information on uninsured/out of network patients, and this may underrepresent the number of patients discharged home from COVID-19 hospitalizations and their post-discharge HCU. Also, our study period spans the first year of the pandemic when many COVID-19 treatments and vaccines were in development or in early stages of use. Therefore, we are unable to determine how many of our patients were vaccinated or had received COVID-19-specific therapy and how these two factors mediate post-discharge HCU. Also, the mean age of our population is 72 years, and our results may be more representative of the older adult population. Similarly, based on limitations from our dataset, we cannot determine the post-discharge mortality rate in our cohort; however, with an estimated 7.8% all-cause post-discharge mortality rate [42] reported in the literature, we know most patients discharged from a COVID-19 hospitalization are alive one year post-discharge [42]. To address this limitation, our cohort only included patients enrolled in insurance/database until the end of the study period. Therefore, each patient contributed person-days until death, loss

of insurance eligibility, or the end of the 9-month follow-up period. Additionally, we acknowledge the multifaceted influences on healthcare utilization, extending beyond direct medical necessity. Integral factors likely include socioeconomic status, health insurance coverage, access to and availability of care, and clinician referrals. We must also consider the potential for bias due to resource exhaustion amid the pandemic's economic fallout. Resource limitation might have prompted a decrease in healthcare visits despite ongoing health impairments, thereby affecting our analysis of utilization rates.

Our study has several strengths including a cohort of patients discharged home obtained from a database that spans nationally. The information obtained from our study provides insights in the HCU trends post-COVID-19 hospitalization in the US. We were also able to follow patients for a considerable period, up to 9 months post discharge, and to compare pre- and post-COVID hospitalization HCU in the same patient population, eliminating potential selection bias or the influence of such confounders as demographic characteristics and prior comorbidities.

## Conclusions and implications

In our nationally representative retrospective study, we identified that HCU remains high among patients discharged to a home setting after a hospitalization due to COVID-19. Health systems and providers may be able to use this information to better deploy resources in the care of this chronically ill population.

## Supporting information

**S1 File. This document contains supplementary tables and figures that provide additional details and analyses supporting the findings of the main manuscript.** The tables and figures included herein offer further insights, data points, and visual representations to enhance the understanding and interpretation of the research presented in the primary manuscript. (DOCX)

## Author Contributions

**Conceptualization:** Daniel Puebla Neira, Gulshan Sharma.

**Data curation:** Efstathia Polychronopoulou, Kuo Yong-Fang.

**Formal analysis:** Efstathia Polychronopoulou, Kuo Yong-Fang.

**Methodology:** Efstathia Polychronopoulou, Kuo Yong-Fang.

**Supervision:** Gulshan Sharma.

**Writing – original draft:** Mohammed Zaidan, Daniel Puebla Neira, Efstathia Polychronopoulou, Kuo Yong-Fang, Gulshan Sharma.

**Writing – review & editing:** Mohammed Zaidan, Daniel Puebla Neira, Efstathia Polychronopoulou, Kuo Yong-Fang, Gulshan Sharma.

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
