## [Decision Letter · Decision Letter 0]

13 Jun 2023

PONE-D-23-10824Health Care Utilization 9 months Pre- and Post- COVID-19 Hospitalization among Patients Discharged AlivePLOS ONE

Dear Dr. Zaidan,

Thank you for submitting your manuscript to PLOS ONE. After careful consideration, we feel that it has merit but does not fully meet PLOS ONE’s publication criteria as it currently stands. Therefore, we invite you to submit a revised version of the manuscript that addresses the points raised during the review process.

The authors employed a nationally-representative database from the United States to examine healthcare utilization patterns up to nine months after patients were discharged home following COVID-19 hospitalization. While the findings demonstrate an increase in healthcare utilization post-discharge for individuals hospitalized due to COVID-19 compared to their pre-pandemic healthcare utilization, the authors must address several crucial concerns before this manuscript can be considered suitable for publication.

As highlighted by the reviewer, it is advisable for the authors to conduct additional analyses comparing healthcare utilization across various sub-groups. The authors must implement sufficient statistical testing in order to substantiate their interpretation of the study's findings. Furthermore, the authors are urged to carefully consider and address the study limitations pointed out by the reviewers.

We look forward to receiving your revised manuscript.

Kind regards,

Raymond Nienchen Kuo, Ph.D

Academic Editor

PLOS ONE

Journal Requirements:

https://www.ncbi.nlm.nih.gov/pmc/articles/PMC9548925/

https://bmcpulmmed.biomedcentral.com/articles/10.1186/s12890-023-02313-x

https://www.jamda.com/article/S1525-8610(21)00762-3/fulltext

In your revision ensure you cite all your sources (including your own works), and quote or rephrase any duplicated text outside the methods section. Further consideration is dependent on these concerns being addressed.

"No"

"No"

Reviewers' comments:

Reviewer's Responses to Questions

**Comments to the Author**

1. Is the manuscript technically sound, and do the data support the conclusions?

Reviewer #1: Yes

Reviewer #2: Yes

2. Has the statistical analysis been performed appropriately and rigorously? 

Reviewer #1: No

Reviewer #2: No

3. Have the authors made all data underlying the findings in their manuscript fully available?

Reviewer #1: Yes

Reviewer #2: Yes

4. Is the manuscript presented in an intelligible fashion and written in standard English?

Reviewer #1: Yes

Reviewer #2: Yes

5. Review Comments to the Author

Reviewer #1: The authors use a large health system in the US to describe patterns in health care use among individuals who were hospitalized with COVID-19 in the early phases of the pandemic, prior to widespread vaccine access or the newer variants. Although they describe and exposure and outcome, formal comparisons are not included. It would be of potential interest if, for example, they compared post-hospitalization use to other conditions (e.g., sepsis, influenza), pre- vs. post-HCU, or even formally assessed trends in HCU over time. At times the authors refer to people who were hospitalized but not discharged home as being excluded, but most of Table 2 describes alternative dispositions for patients who were not discharged home. Perhaps another formal comparison would be of interest in that regard, e.g., predictors of home vs. not-home discharge.

It is important to note in more detail that health care use in the US is not driven solely by patient need for medical care. It involves a complex interplay of multiple factors, including health insurance, access to care, available medical care, and referrals. Patients cannot access medical care they (or their clinicians) do not know they would benefit from (e.g., increasing evidence for increased risk of diabetes following infection suggests some patients might benefit from endocrinology evaluation; GI complications are also apparent, as are fertility issues, but few were aware of these issues at the time of the study); second, patients who have exhausted available resources do eventually stop going to as many medical appointments, even in cases where their health remains severely impaired. Please comment.

With regards to healthcare encounters, is it possible to include home health encounters?

Did the authors consider looking for potential effect modification by sex? Or by ICU admission during the index hospitalization? What % of cases were re-infections, and if an important %, what if any difference is there in HCU by reinfection?

The authors describe covariates, although no adjusted models were run. Perhaps baseline patient characteristics might a more accurate term. With regards to these variables, why gender (instead of sex), what were the race categories (and were they patient reported or something else), and what about other important potential confounders, e.g., education, insurance, occupation, number o vaccinations? Finally, inclusion of comorbid conditions only in the prior 12 months likely significantly underestimated true co-morbid conditions, is it possible to expand the look-back window to several years?

Table 1: What % were covered by commercial vs. other health insurance?

Table 2 seems more appropriate for study flow- these patients were excluded from the cohort.

There are more limitations than the authors list, including the following:

- Use of only patients insured by a large commercial company or Medicare Advantage with at least 2 months of pre-COVID data means the study population includes people who have relatively good access to medical care. The authors do not report what % had commercial insurance, but those people would have been healthy (and young enough) to have employment. How representative is this of the overall population?

- What about care people may have received in other locations, e.g., out of network?

- The study time is limited to largely pre-vaccination stages of the pandemic, and during early variants, when monoclonals were available and used more widely – would these findings be generalizable or informative now?

Line 181 – There is an interesting but unfounded (by reproducible studies) assumption that patients are ‘merely anxious or depressed’ after SARS-CoV-2 infection, implying they should be able to use CBT or positive self-talk to resolve their symptoms. In an effort to discourage these and other similar implications, please remove the phrase “decline in mental health” from line 181 and instead simply report the numbers.

Reviewer #2: This study provides evidence of healthcare utilization (HCU) trends of COVID-19 patients 9 months post-diagnosis using a sample of the U.S. COVID-19 patient population. The cohort creation process is transparent and the paper is well written. However, the current analysis is not sufficient to merit publication at PLOS ONE. First, it is not entirely clear what is the biggest contribution of the study. Given that there are other studies that extend the time horizon of long COVID HCU to twelve months (Roth et al., 2022), the authors should argue how these results may be more informative to other longitudinal studies of HCU and long COVID on the basis of the quality of their analysis, as opposed to the long time span of the COVID-19 post-diagnosis period only. It is not surprising that HCU rises post-diagnosis; the authors could also try to timestamp when this increased HCU subsides for different age groups (Koumpias et al., 2022).

The authors should consider these easily executable revisions below to enhance the rigor of the analytical approach.

Major comments:

The biggest concern is the lack of any account of patient insurance status. The confluence of Medicare-eligible and Medicare non-eligible population in the study cohort leads to overestimation of the influence of a COVID-19 diagnosis on HCU because the latter group (<65years old) is less likely to engage in HCU due to relatively higher costs they may face. Therefore, the authors should report HCU separately for commercially-insured (<65yrs old) and Medicare/Medicare Advantage patients.

It is also mentioned that this is a descriptive analysis; yet no statistical tests are being used, whatsoever. At a minimum, the authors shoould report whether pre-, post-diagnosis levels are statistically different using t-tests. For instance, the first sentence of the introduction reads: "In our nationally-representative study of patients discharged home after a COVID-19 hospitalization, we found that HCU remained significantly high 9 months after discharge from index hospitalization." It appears highly likely that the increase in HCU is detectable at conventional levels of statistical significance, too. After conducting the statistical analysis, this statement could be revised to explicitly state whether increased HCU remained higher in a statistically significant way.

Another concern is that HCU is driven by intensity of hospitalization which varies in a way that the authors do not sufficiently account for. Separating the results by specialty is certainly a step towards that direction. It would be most informative to show how the trends of post-diagnosis HCU for different subgroups of varying hospitalization intensity. In fact, the authors should consider shedding light to any differential HCU responses by LOS. (e.g. by LOS quartiles). This would make a genuine contribution by further illustrating post-discharge differences in HCU based on severity of COVID-19 infection. However, the authors do not mention changes in patient LOS until page 9. Given that there is relatively less evidence on LOS changes than HCU changes following COVID-19 diagnosis, this novel finding should be discussed earlier if not employed as another variable in cross-tabulations of HCU pre- and post-diagnosis.

Minor comments:

Another source of measurement error causing underestimation of the association of COVID-19 with post-diagnosis HCU is due to sample attrition from death or loss of insurance. To examine the sensitivity of their findings to this potential issue, the authors should use a balanced panel of patients who contribute person-days both pre-diagnosis and during the third post-diagnosis time interval (6-9) as a supplementary robustness check.

Related, the authors should explore whether information regarding COVID-19 diagnosis is available in the secondary and subsequent diagnosis lines and to what extent this leads to significant undercount of COVID-19 patients. It would be very helpful to complement the results a discussion of the frequency of COVID-19 diagnosis non-primary line reporting.

Re: Data Availability - Unsure whether data is available after all given that this is the proprietary OptumInsights de-identified Clinformatics Data Mart. This may need to be corrected.

Finally, are the results robust to the inclusion of the outcome in its raw form, measured in levels? It would be useful to show whether the transformation to person-days has any influence on the results.

A quick literature review identified the following journal articles pertinent to this study:

References:

- Koumpias AM, Schwartzman D, Fleming O. Long-haul COVID: healthcare utilization and medical expenditures 6 months post-diagnosis. BMC Health Services Research. 2022 Aug 8;22(1):1010.

- Roth SE, Govier DJ, Marsi K, Cohen-Cline H. Differences in outpatient health care utilization 12 months after COVID-19 infection by race/ethnicity and community social vulnerability. International Journal of Environmental Research and Public Health. 2022 Mar 15;19(6):3481.

- Zhou X, Andes LJ, Rolka DB, Imperatore G. Changes in health care utilization among Medicare beneficiaries with diabetes two years into the COVID-19 pandemic. Ajpm Focus. 2023 Jun 1:100117.

6. PLOS authors have the option to publish the peer review history of their article (what does this mean?). If published, this will include your full peer review and any attached files.

Reviewer #1: No

Reviewer #2: No

---

## [Author Response · Author response to Decision Letter 0]

12 Sep 2023

Thank you for the constructive feedback. Please see the "response to reviewers" document for our full responses.

---

## [Decision Letter · Decision Letter 1]

16 Oct 2023

PONE-D-23-10824R1Health Care Utilization 9 months Pre- and Post- COVID-19 Hospitalization among Patients Discharged AlivePLOS ONE

Dear Dr. Zaidan,

Thank you for submitting your manuscript to PLOS ONE. After careful consideration, we feel that it has merit but does not fully meet PLOS ONE’s publication criteria as it currently stands. Therefore, we invite you to submit a revised version of the manuscript that addresses the points raised during the review process.

We look forward to receiving your revised manuscript.

Kind regards,

Raymond Nienchen Kuo, Ph.D

Academic Editor

PLOS ONE

Journal Requirements:

Additional Editor Comments:

We acknowledge the effort and dedication you have exhibited in responding to the initial reviews and improving your manuscript. However, it has come to our attention that there are still several significant issues raised by Reviewer #1 that necessitate further elucidation. One such area pertains to how this study could potentially bridge the existing knowledge gap related to inpatient care related to COVID-19 infection. It would be constructive to elaborate on this aspect in your manuscript.

In addition, we suggest that you provide a concise description of the data source that was used in your research. It would also be beneficial to discuss the potential impact of your sample's characteristics on the generalizability of your study findings. This will allow readers to understand the context and implications of your research better.

Reviewers' comments:

Reviewer's Responses to Questions

**Comments to the Author**

1. If the authors have adequately addressed your comments raised in a previous round of review and you feel that this manuscript is now acceptable for publication, you may indicate that here to bypass the “Comments to the Author” section, enter your conflict of interest statement in the “Confidential to Editor” section, and submit your "Accept" recommendation.

Reviewer #1: (No Response)

Reviewer #2: All comments have been addressed

2. Is the manuscript technically sound, and do the data support the conclusions?

Reviewer #1: Partly

Reviewer #2: Yes

3. Has the statistical analysis been performed appropriately and rigorously? 

Reviewer #1: No

Reviewer #2: Yes

4. Have the authors made all data underlying the findings in their manuscript fully available?

Reviewer #1: No

Reviewer #2: No

5. Is the manuscript presented in an intelligible fashion and written in standard English?

Reviewer #1: Yes

Reviewer #2: Yes

6. Review Comments to the Author

Reviewer #1: The authors have responded to some questions and comments to the best of their ability, and comparison of 9 months pre- vs. post HCU hospitalization for COVID was added, which his helpful. However, there remain significant limitations within the available data that perhaps cannot be addressed. Overall, however, it is not surprising that people who were hospitalized required more health care after hospitalization and they did poorly in general, even among those who were only eligible because they survived at least a year after hospital discharge. One key question that this study cannot answer is whether or to what degree hospitalization for COVID may differ from other causes of hospitalization.

Can the authors clearly explain what this study adds beyond a prior publication (https://www.ncbi.nlm.nih.gov/pmc/articles/PMC9548925/) how it adds materially to our understanding of health care use post hospitalization for COVID (specifically, compared to other reasons for hospitalization).

Otherwise, I have the following additional comments/questions:

Please provide a brief description of the kinds of patients who would be included in Optum’s Clinformatics Data Mart – what % of the US population are generally included? Are these people generally of similar, worse, or better health than the general population? What kinds of medical centers contribute data, e.g., typically community vs. academic? What % are rural?

Readers have a general idea of, for example, the patient population and generalizability for VHA studies, but that is not the case here. For provide a similar summary for patients in this cohort.

Page 7 line 127 – Please clarify that this was the primary admission diagnosis (In Figure 1, I see that this appears to be a hospital admission diagnosis, is that correct?)

In the discussion/limitations, please include the risk of introducing survivor bias by requiring that people have at least 12 months of follow-up. Why not allow variable follow-up and simply account for it in analysis, and not risk survivor bias?

What are the test characteristics for the ICD code for accurate case identification? (e.g., sensitivity, specificity, PPV, NPV)

I initially misread Figure to read that 63,161 people were excluded due to loss of insurance prior to hospital discharge. Please restructure Figure 1 to list the N excluded (%), similar to reporting for clinical trials.

Table 1: Lack of insurance information is an important limitation. How were the comorbid conditions identified (i.e., in a supplement, include ICD codes, etc)

Table 2: Please move the units for each measure of HCU to the row label to make them more clear and please provide more explanation of the measures of health care utilization. E.g., “ED visits per 10,000 person-days”. Are the reported pre- vs. post-hospitalization HCU reflective of the mean or median or some other measure? Please include the appropriate measure of variability around that reported summary (e.g., SD, IQR, etc). Is hospital LOS per capita as well? Why was 9 months before and after chosen? Why not a year? (A year would be easier to compare to other information, e.g., the baseline proportion of people with an ED visit per year, to assess generalizability)

Please confirm that post hospitalization HCU measures do not include the index hospitalization.

Tables 1, 2, and 3 are currently in the Methods section, but shouldn’t they be referred to and located in the Results section?

Results:

Why might people who were in the ICU have shorter length of stay? (How does that translate into a RR>1? Am I interpreting that result incorrectly?)

Discussion:

For people in other countries, please provide some context for health care utilization. In general, how much might an outpatient visit cost, out of pocket? What % of a hospitalization might patients have to pay out of pocket, and approximately how much might that be in absolute USD? This provides context for what might drive patient behavior, specifically health-seeking behaviors. Might some patients avoid outpatient care if it cost $25/visit, while an ED visit or in-patient hospitalization might have $0 out of pocket costs?

Is it possible to include any information about why patients were re-hospitalized? E.g., admission for ACS, neurological, PNA, dyspnea, or other causes?

For consistency, I suggest using the terms male and female throughout, since sex was likely the measured variable in available data, not gender.

Reviewer #2: (No Response)

7. PLOS authors have the option to publish the peer review history of their article (what does this mean?). If published, this will include your full peer review and any attached files.

Reviewer #1: No

Reviewer #2: No

---

## [Author Response · Author response to Decision Letter 1]

5 Dec 2023

Reviewers' comments: 

Reviewer's Responses to Questions 

Comments to the Author 

1. If the authors have adequately addressed your comments raised in a previous round of review and you feel that this manuscript is now acceptable for publication, you may indicate that here to bypass the “Comments to the Author” section, enter your conflict of interest statement in the “Confidential to Editor” section, and submit your "Accept" recommendation. 

Reviewer #1: (No Response) 

Reviewer #2: All comments have been addressed 

2. Is the manuscript technically sound, and do the data support the conclusions? 

Reviewer #1: Partly 

Reviewer #2: Yes 

3. Has the statistical analysis been performed appropriately and rigorously? 

Reviewer #1: No 

Reviewer #2: Yes 

4. Have the authors made all data underlying the findings in their manuscript fully available? 

Reviewer #1: No 

Reviewer #2: No 

5. Is the manuscript presented in an intelligible fashion and written in standard English? 

Reviewer #1: Yes 

Reviewer #2: Yes 

6. Review Comments to the Author 

Reviewer #1: 

The authors have responded to some questions and comments to the best of their ability, and comparison of 9 months pre- vs. post HCU hospitalization for COVID was added, which his helpful. However, there remain significant limitations within the available data that perhaps cannot be addressed. Overall, however, it is not surprising that people who were hospitalized required more health care after hospitalization and they did poorly in general, even among those who were only eligible because they survived at least a year after hospital discharge. 

R1: 

One key question that this study cannot answer is whether or to what degree hospitalization for COVID may differ from other causes of hospitalization. 

A1: 

Thank you for the criticism. Our main purpose of this article was to provide a descriptive analysis of health care utilization of patients post hospitalization for COVID pneumonia. We did not intend to compare how hospitalization for COVID pneumonia may differ from other pneumonias or other causes. Other published works have made such comparisons, such as these 2 articles. 

COVID pneumonia hospitalization had increased pulmonary shunts that is not commonly seen in other types of pneumonia. (Novelli et al) Another article noted that patients with COVID pneumonia who required mechanical ventilation had longer times to liberation from mechanical ventilation. (Nolley et al). 

Novelli, Malandrino, Balbi, et al. Shunt fraction and radiological involvement in Covid-19 related Acute Respiratory Failure. European Respiratory Journal Sep 2023, 62 (suppl 67) PA5105; DOI: 10.1183/13993003.congress-2023.PA5105 

Nolley EP, Sahetya SK, Hochberg CH, et al. Outcomes Among Mechanically Ventilated Patients With Severe Pneumonia and Acute Hypoxemic Respiratory Failure From SARS-CoV-2 and Other Etiologies. JAMA Netw Open. 2023;6(1):e2250401. doi:10.1001/jamanetworkopen.2022.50401 

R2: 

Can the authors clearly explain what this study adds beyond a prior publication (https://www.ncbi.nlm.nih.gov/pmc/articles/PMC9548925/) how it adds materially to our understanding of health care use post hospitalization for COVID (specifically, compared to other reasons for hospitalization). 

A2: 

Thank you for pointing out this prior publication. This was our poster publication that was presented at the CHEST conference 2022 and published online. Since this abstract, we have performed a statistical analysis and sensitivity analysis on healthcare utilization post discharge the index hospitalization. 

Previous studies compared HCU between patients with and without COVID. 

In this article, we did within subject comparison (before and after COVID for each patient). We also compared the within subject changes by gender and ICU. We did not intend to compare how hospitalization for COVID pneumonia may differ from other pneumonias or other causes. 

R3: 

Otherwise, I have the following 3 additional comments/questions: 

Please provide a brief description of the kinds of patients who would be included in Optum’s Clinformatics Data Mart – what % of the US population are generally included? 

Are these people generally of similar, worse, or better health than the general population? 

What kinds of medical centers contribute data, e.g., typically community vs. academic? What % are rural? 

Readers have a general idea of, for example, the patient population and generalizability for VHA studies, but that is not the case here. For provide a similar summary for patients in this cohort. 

A3: 

Optum’s Clinformatics Data Mart contains claims from enrollees with commercial insurance or Medicare Advantage Plans. The database comprises of over 63 million unique enrollees in the United States from all states. 

The majority (>= 95%) of the enrollees in Optum’s CDM have commercial insurance. 

Traditional Medicare or Medicaid enrollees are not included in the database, so it is likely that the dataset is less representative of the older or low socioeconomic status population. 

Because the data in Optum’s CDM come from insurance claims, there are no limitations on the type of medical center that can contribute data (e.g., rural / urban or academic/community). 

When a CDM patient files a claim with their insurance, the data are captured, regardless of the medical center/office he visited. 

We have added these points to the Data Source paragraph in the methods section 

R4: 

Page 7 line 127 – Please clarify that this was the primary admission diagnosis (In Figure 1, I see that this appears to be a hospital admission diagnosis, is that correct?) 

A4: 

Yes. This was the primary admission diagnosis for the hospitalization. 

R5: 

In the discussion/limitations, please include the risk of introducing survivor bias by requiring that people have at least 12 months of follow-up. Why not allow variable follow-up and simply account for it in analysis, and not risk survivor bias? 

A5: 

Thank you for your comment. To address the concern of survivor bias, we required participants to have at least 12 months of data prior to their primary COVID-related hospitalization for accurate comorbidity assessment and utilized a person-days approach to accommodate variable follow-up durations. We did not require follow-up AFTER hospitalization. We used person-days to capture to allow for variable follow up and limit survivor bias. 

R5: 

What are the test characteristics for the ICD code for accurate case identification? (e.g., sensitivity, specificity, PPV, NPV) 

A5: 

While there are no published studies on ICD code accuracy using Optum’s CDM specifically, a study (Kadri et. Al) using the same ICD-codes for identification of COVID-19 disease in another insurance claims database found the following: "sensitivity was 98.01% (95% CI, 97.62%-98.39%), the specificity was 99.04% (95% CI, 98.95%-99.13%), the positive predictive value was 91.52% (95% CI, 90.77%-92.27%), and the negative predictive value was 99.79% (95% CI, 99.75%-99.83%).” 

Kadri SS, Gundrum J, Warner S, et al. Uptake and Accuracy of the Diagnosis Code for COVID-19 Among US Hospitalizations. JAMA. 2020;324(24):2553–2554. doi:10.1001/jama.2020.20323 

R6: 

I initially misread Figure to read that 63,161 people were excluded due to loss of insurance prior to hospital discharge. Please restructure Figure 1 to list the N excluded (%), similar to reporting for clinical trials. 

A6: 

Thank you for the suggestion. We have made this change. 

Please see the new Figure 1 

R7: 

Table 1: Lack of insurance information is an important limitation. How were the comorbid conditions identified (i.e., in a supplement, include ICD codes, etc) 

A7: 

All patients had insurance under Optum’s insurance contractor (large national commercial insurance or Medicare Advantage). Over 95% had commercial insurance and a little over 4% had Medicare advantage (we unfortunately do not have more specific information on insurance for each patient). We required all patients covered by Optum insurance contracts in the 12 months before hospitalization in order to identify comorbidity. 

Additonally, we have added this table in the supplement section 

Disease 

ICD-10 codes 

COVID-19 

U07.1 

Diabetes 

E10.xx, E11.xx, E13.xx 

Hypertension 

I10.x, I11.xx, I12.xx, I13.xx I15.xx, I67.4 

Asthma 

J45.xx 

COPD 

J41.8, J42.xx, J43.xx, J44.xx 

CKD 

N18.9 

ESRD 

N18.6 

Stroke 

I63.xx, I64.xx, I69.3x, G45.9 

Heart disease 

I09.9, I11.0, I13.xx, I25.5, I42.xx, I43.xx, I50.xx 

Cancer 

C00.x – C96.x 

CAD 

I25.1X 

Liver disease 

K70 – K74, K76.xx 

R8: 

Table 2: Please move the units for each measure of HCU to the row label to make them more clear and please provide more explanation of the measures of health care utilization. E.g., “ED visits per 10,000 person-days”. 

A8: 

Thank you for this suggestion. Completed. 

R9: 

Are the reported pre- vs. post-hospitalization HCU reflective of the mean or median or some other measure? 

A9: 

We report healthcare use (e.g. number of visits) as incidence per 10,000 person – days. It is an epidemiologic measure, which accounts for the actual time each participant contributes to the study. 

We mention this on lines 147-150 and each table has this mentioned at the bottom. 

R10: 

Please include the appropriate measure of variability around that reported summary (e.g., SD, IQR, etc). Is hospital LOS per capita as well? 

A10: 

There is no measure of variability for incidence, other than the reported risk ratio when comparing two incidence rates. We reported LOS as total days of inpatient stay per 10,000 person-days. 

R11: 

Why was 9 months before and after chosen? Why not a year? (A year would be easier to compare to other information, e.g., the baseline proportion of people with an ED visit per year, to assess generalizability) 

A11: 

During the time of our initial data acquisition, we didn't have data for 12 months post-COVID hospitalization. 

R12: 

Please confirm that post hospitalization HCU measures do not include the index hospitalization. 

A12: 

This is correct. All the HCU measures DID NOT include the index hospitalization. 

R13: 

Tables 1, 2, and 3 are currently in the Methods section, but shouldn’t they be referred to and located in the Results section? 

A13: 

Thank you for the suggestion, the tables have been moved to the results section. 

R14: 

Why might people who were in the ICU have shorter length of stay? (How does that translate into a RR>1? Am I interpreting that result incorrectly?) 

A14: 

Thank you for your comment. We found that increase of covid HCU from pre to post period in patients with ICU stays was larger than those without, hence the RR>1. Those patients with ICU stays during COVID hospitalization had much large increase of length of stay than those without. 

Discussion: 

R15: 

For people in other countries, please provide some context for health care utilization. In general, how much might an outpatient visit cost, out of pocket? 

A15: 

Ambulatory care, visit complexity is stratified from Level 1 for minor issues, costing an average of $46, to Level 5 for the most severe cases at $182 per visit. Level 3, the median for common visits, averages $90. The overall outpatient visit cost in 2018 averaged $105. 

From 2008 to 2018, the cost for Level 1 visits rose by $15 (52%), while Level 5 visits increased by $49 (37%). Level 3 visits, the most frequent, saw a $20 rise (29%), indicating a notable trend in healthcare expenditure growth. 

"How costly are common health services in the United States?" Peterson-KFF Health System Tracker. https://www.healthsystemtracker.org/chart-collection/how-costly-are-common-health-services-in-the-united-states/#item-start. Accessed November 7, 2023.

R16: 

What % of a hospitalization might patients have to pay out of pocket, and approximately how much might that be in absolute USD? This provides context for what might drive patient behavior, specifically health-seeking behaviors. Might some patients avoid outpatient care if it cost $25/visit, while an ED visit or in-patient hospitalization might have $0 out of pocket costs? 

A16: 

Hospitalization costs incur varied out-of-pocket expenses, influenced by insurance coverage. In 2020, average out-of-pocket costs for COVID-19 hospitalizations were approximately $1,653 under traditional plans and $1,961 for consumer-driven plans. A related study indicated a median cost for privately insured patients ranging from $59 to $842 due to COVID-19. 

"Assessment of Out-of-Pocket Spending for COVID-19 Hospitalizations in the US in 2020." JAMA Network. https://jamanetwork.com/journals/jama/fullarticle/2780416. Accessed November 7, 2023. 

On a broader note, U.S. employees paid an average of $1,763 out-of-pocket before reaching their insurance deductibles last year. Such costs are substantial and may affect decisions regarding follow-up outpatient care, in contrast to some plans covering emergency or inpatient services fully, thereby incurring no out-of-pocket expenses 

"Average out-of-pocket healthcare costs." Bankrate. https://www.bankrate.com/insurance/health/average-out-of-pocket-healthcare-costs/

R17: 

Is it possible to include any information about why patients were re-hospitalized? E.g., admission for ACS, neurological, PNA, dyspnea, or other causes? 

A17: 

Unfortunately, our contract of the Optum data set was expired on June 2023 and this is not possible. 

R18: 

For consistency, I suggest using the terms male and female throughout, since sex was likely the measured variable in available data, not gender. 

A18: 

Thank you for the suggestions. We have made the appropriate changes. 

Reviewer #2: (No Response) 

7. PLOS authors have the option to publish the peer review history of their article (what does this mean?). If published, this will include your full peer review and any attached files. 

Do you want your identity to be public for this peer review? For information about this choice, including consent withdrawal, please see our Privacy Policy. 

Reviewer #1: No 

Reviewer #2: No 

---

## [Decision Letter · Decision Letter 2]

12 Feb 2024

PONE-D-23-10824R2Health Care Utilization 9 months Pre- and Post- COVID-19 Hospitalization among Patients Discharged AlivePLOS ONE

Dear Dr. Zaidan,

Thank you for submitting your manuscript to PLOS ONE. After careful consideration, we feel that it has merit but does not fully meet PLOS ONE’s publication criteria as it currently stands. Therefore, we invite you to submit a revised version of the manuscript that addresses the points raised during the review process.

We look forward to receiving your revised manuscript.

Kind regards,

Academic Editor

PLOS ONE

**Additional Editor Comments:**

Please revise.

Reviewers' comments:

Reviewer's Responses to Questions

**Comments to the Author**

1. If the authors have adequately addressed your comments raised in a previous round of review and you feel that this manuscript is now acceptable for publication, you may indicate that here to bypass the “Comments to the Author” section, enter your conflict of interest statement in the “Confidential to Editor” section, and submit your "Accept" recommendation.

Reviewer #2: All comments have been addressed

Reviewer #3: (No Response)

2. Is the manuscript technically sound, and do the data support the conclusions?

Reviewer #2: Yes

Reviewer #3: Partly

3. Has the statistical analysis been performed appropriately and rigorously? 

Reviewer #2: Yes

Reviewer #3: Yes

4. Have the authors made all data underlying the findings in their manuscript fully available?

Reviewer #2: No

Reviewer #3: No

5. Is the manuscript presented in an intelligible fashion and written in standard English?

Reviewer #2: Yes

Reviewer #3: Yes

6. Review Comments to the Author

Reviewer #2: (No Response)

Reviewer #3: This study by Zaidan et al, described the increase in Health Care Utilization (HCU) in patients post COVID-19 hospital discharge (follow-up = 9 months). The manuscript is well written and includes appropriate analyses and tables.

The tables should be referred to in the results & not the methods.

They also have numerous subscripts within the title of the tables that are unnecessary. Just need a table legend without the addition of subscripts within the title (Minor point).

I am slightly confused about the inclusion/exclusion criteria. In the "Cohort Selection" section of the methods, the authors state that "Our samples included all adults hospitalized with primary diagnosis of COVID-19 who were discharged home (with or without HOME HEALTH care)..." However, in Figure 1 they state that "Not discharged home or with HOME HEALTH" was part of the exclusion criteria. This needs to be clarified, were those with HOME HEALTH included of excluded. This may have an impact on the amount of HCU. If HOME HEALTH participants were included then this would likely have an impact on HCU.

Similarly, another potentially observation that could inform future pandemic preparedness would be analysis on whether particular co-morbidities were associated with increased HCU post-COVID discharge.

Please define OFFICE (mentioned in the results (e.g. p13; line 232, p14; line 248) which was not described within the results.

A limitation of this work is perhaps the limited ability to extrapolate these findings to a larger cohort (throughout the USA), given that this cohort is likely to have a higher socioeconomic status (given the degree of insurance cover) and are of older age (72 years). If the authors were to divide the cohort by age (in a sub analysis) this would add more weight to generalizing to a larger population.

7. PLOS authors have the option to publish the peer review history of their article (what does this mean?). If published, this will include your full peer review and any attached files.

Reviewer #2: No

Reviewer #3: No

---

## [Author Response · Author response to Decision Letter 2]

11 Apr 2024

Reviewers:

We note that the grant information you provided in the ‘Funding Information’ and ‘Financial Disclosure’ sections do not match.

When you resubmit, please ensure that you provide the correct grant numbers for the awards you received for your study in your cover letter; we will change the online submission form on your behalf.

Response: 

I spoke to Dr. Kuo and she reports that that her grants are related to opioid prescription, cancer, and ADRD. None of these grants are related to this study.

So I will remove her grants from the page and resubmit. 

Reviewer #3:  

Comment: The tables should be referred to in the results & not the methods. They also have numerous subscripts within the title of the tables that are unnecessary. Just need a table legend without the addition of subscripts within the title (Minor point). 

Response: Thank you for your comment. All tables references have been removed from the methods. Tables are referenced in the results section. Also, subscripts from the titles have been removed and the table legends summarized. 

Comment: I am slightly confused about the inclusion/exclusion criteria. In the "Cohort Selection" section of the methods, the authors state that "Our samples included all adults hospitalized with primary diagnosis of COVID-19 who were discharged home (with or without HOME HEALTH care)..." However, in Figure 1 they state that "Not discharged home or with HOME HEALTH" was part of the exclusion criteria. This needs to be clarified, were those with HOME HEALTH included of excluded. This may have an impact on the amount of HCU. If HOME HEALTH participants were included then this would likely have an impact on HCU. 

Response: Thank you for your comment. We agree with the reviewer. The way figure one was written lead to potential confusion. To clarify, our study only included patients who were discharged home (with or without home health). So, if a patient was not discharged home, then it was excluded. Figure one has been edited to clarify this point. 

Comment: Similarly, another potentially observation that could inform future pandemic preparedness would be analysis on whether particular co-morbidities were associated with increased HCU post-COVID discharge. 

Response: PLEASE RESPOND 

Comment: Please define OFFICE (mentioned in the results (e.g. p13; line 232, p14; line 248) which was not described within the results. 

Response: Thank you for your comment. We have added the definition of office in the methods section. A copy of the paragraph follows “. Based on billing information, we examined outpatient visits (from here on out referred to as office visits) to primary care providers, including family medicine, internal medicine, and nurse practitioner visits, and certain sub-specialties, such as cardiology, pulmonary medicine, endocrinology, neurology, physical medicine and rehabilitation, psychiatry, and other mental health professionals” 

Comment: A limitation of this work is perhaps the limited ability to extrapolate these findings to a larger cohort (throughout the USA), given that this cohort is likely to have a higher socioeconomic status (given the degree of insurance cover) and are of older age (72 years). If the authors were to divide the cohort by age (in a sub analysis) this would add more weight to generalizing to a larger population. 

Response: Thank you for your comment. Socioeconomic status has been added as a limitation. Please see discussion section under limitations. 

Please see the table below. We have made a table that divides the cohort by age. This has been added to the supplemental tables.

---

## [Decision Letter · Decision Letter 3]

26 Apr 2024

Health Care Utilization 9 months Pre- and Post- COVID-19 Hospitalization among Patients Discharged Alive

PONE-D-23-10824R3

Dear Dr. Zaidan,

We’re pleased to inform you that your manuscript has been judged scientifically suitable for publication and will be formally accepted for publication once it meets all outstanding technical requirements.

Kind regards,

Academic Editor

PLOS ONE

Additional Editor Comments (optional):

Reviewers' comments:

Reviewer's Responses to Questions

**Comments to the Author**

1. If the authors have adequately addressed your comments raised in a previous round of review and you feel that this manuscript is now acceptable for publication, you may indicate that here to bypass the “Comments to the Author” section, enter your conflict of interest statement in the “Confidential to Editor” section, and submit your "Accept" recommendation.

Reviewer #2: All comments have been addressed

Reviewer #3: (No Response)

2. Is the manuscript technically sound, and do the data support the conclusions?

Reviewer #2: Yes

Reviewer #3: Yes

3. Has the statistical analysis been performed appropriately and rigorously? 

Reviewer #2: Yes

Reviewer #3: Yes

4. Have the authors made all data underlying the findings in their manuscript fully available?

Reviewer #2: No

Reviewer #3: No

5. Is the manuscript presented in an intelligible fashion and written in standard English?

Reviewer #2: Yes

Reviewer #3: Yes

6. Review Comments to the Author

Reviewer #2: (No Response)

Reviewer #3: (No Response)

7. PLOS authors have the option to publish the peer review history of their article (what does this mean?). If published, this will include your full peer review and any attached files.

Reviewer #2: **Yes: **Antonios Marios Koumpias

Reviewer #3: No

---

## [Editor Report · Acceptance letter]

27 May 2024

PONE-D-23-10824R3 

PLOS ONE

Dear Dr. Zaidan, 

I'm pleased to inform you that your manuscript has been deemed suitable for publication in PLOS ONE. Congratulations! Your manuscript is now being handed over to our production team.

Kind regards, 

on behalf of

Dr. Robert Jeenchen Chen 

Academic Editor

PLOS ONE